# Exploring Effective Physics Teaching Strategies in High Schools during the COVID-19 Pandemic

**Roberto Luca Mazzola** [1,*] **, Paolo Gondoni** [2] **, Matteo Bozzi** [1] **, Juliana Elisa Raffaghelli** [3] **and Maurizio Zani** [1]

1 Department of Physics, Politecnico di Milano, 20133 Milan, Italy; matteo.bozzi@polimi.it (M.B.); maurizio.zani@polimi.it (M.Z.)
2 Istituto Superiore Ten.Vasc. A. Badoni, 23900 Lecco, Italy; paolo.gondoni@iisbadoni.edu.it
3 Department of Philosophy, Sociology, Education and Applied Psychology (FISPPA), University of Padua, 35122 Padova, Italy; juliana.raffaghelli@unipd.it
* Correspondence: roberto.mazzola@polimi.it

**Abstract:** The need for educational approaches that comply with the restrictions arising from the COVID-19 pandemic has raised a number of critical issues for students of different age groups. The delicate transition between high school and university has become a key point to focus on, leading many institutions to replan projects dedicated to students involved in this transition. A Physics vocational training project for high school students was carried out in the school year 2020–2021, and it was replicated in the school year 2021–2022. The project included webinars, self-assembled laboratory group experiences, and peer evaluation. The starting point on which we designed our project is that learning is an experience; thus, we built the entire project by particularly focusing on two peculiarities. One peculiarity is the assessment methods: student presentations describing their own experiences were evaluated by teachers and their peers. The second peculiarity is the open approach with respect to how students handle experimental activities. We present a description of these projects along with the results of an evaluation survey filled out by the participants and a descriptive analysis of the assessment strategies. Students appreciated the design of the entire project and, better still, the peer evaluation process. Moreover, we discovered that the evaluation provided by the teachers is lower compared to the assessment reported by the students. This disparity holds potential significance from a statistical perspective and warrants further investigation.

**Keywords:** peer evaluation; inquiry-based learning; teaching strategies; experimental work; high school project

## 1. Introduction

Over the last two years, the world of education has been called upon to cope with unprecedented challenges as a consequence of the SARS-CoV-2 pandemic [1]. Schools and universities were abruptly forced to abandon in-presence activities [2], and these institutions were suddenly unable to use the tools they mainly relied upon in the past; there was a need to turn to remote-based virtual forms of teaching [3–5]. Students in the 17–19 age range, while suffering from a wide range of limitations in the COVID-19 era, still had to be accompanied during their delicate transition from high school to either university [6] or employment. Such a transition requires a thorough thinking process that is based on an enhancement of individual competencies, and adequate support should be provided for those who enroll as freshmen in universities in order to prevent early dropouts.

### 1.1. The Vocational Training Programs in Italy

The literature [7] suggests that vocational training programs have positive effects on youth employment outcomes, reduce the use of informal job search channels, and improve skill matching, especially in the central–northern regions of Italy. The Italian

educational system has introduced an experimental vocational training course—Percorsi per le Competenze Trasversali e l'Orientamento system (PCTO), defined as "Paths for Transversal Skills and School Guidance" [8]—which is aimed at helping students who approach the world of employment [9]. During their last three years of high school, students are employed in a vocational training program at different public or private institutions, factories, research centers, firms, universities, etc. High schools, especially, have the opportunity to structure their PCTO into three modules, each with different objectives and activities, both at school and in work settings. This approach aims to meet the diverse needs of students in terms of curriculum and career guidance. The three modules, namely "Me and Work", "Experimenting with Myself", and "Making Decisions" amount to a total of 90 h. In the second module, students are exposed to various work environments. The activities in this module focus on enhancing their skills in information utilization, decision-making, project development, and managing challenges associated with their chosen paths. The specific objectives of this phase include:

○ Operating responsibly and taking initiative in diverse cultural, corporate, and organizational contexts.
○ Collaborating effectively in group settings.
○ Acquiring and developing knowledge and skills to construct projects.
○ Planning action strategies and finding solutions to complete projects or assigned tasks successfully.
○ Evaluating their own attitudes, knowledge, and abilities in relation to the requirements of higher education and the job market.
○ Strengthening their ability to guide themselves and make independent choices.

Engaging with universities [6], institutions, and companies through educational experiences, work projects, and internships creates a knowledge laboratory where students can cultivate critical thinking, creativity, and organizational skills. By participating in alternative teaching experiences with diverse models and strategies, students can apply what they have learned in the classroom

### 1.2. The Vocational Training Project We Designed

In this framework, we present the activities that were developed as PCTO and performed in the school year 2020–2021 and in the school year 2021–2022. These activities were proposed to the students in their next-to-last (4th year out of 5 years) high school year. It is worth noting that they are limited to the field of Physics but are extendable in principle to other disciplines. The activity that we proposed as PCTO, initially starting from a series of webinars about optics and different perspectives on the experimental method, led different groups of students (three or four students for each group) to devise an acoustic interferometry experiment from scratch. Finally, they presented their work and evaluated the presentations of their classmates. According to Biggs [10], learning is not something granted or transmitted from the teacher to the student, but rather learning experiences must be created by the students. In this constructivist approach, which is a prominent pedagogical theory rooted in the cognitive perspective, knowledge is not something that is given but is rather constructed by individuals. This construction occurs due to the internal processing of emotions, prior knowledge, values, and belief systems. Learning, therefore, emerges from students engaging in recursive processes that involve experiences, abstractions, inference, problem-solving, and the recombination of information. Prominent figures in the field of pedagogy, such as Piaget, Bloom, Bruner, Kolb, Kelly, and Montessori, provided significant contributions to this theory. Thus, we attempted to transform traditional teaching practices into processes centered around the student. The entire project was built by focusing on the following three fundamental rules of learning [11]:

○ Learning is an experience and not a download process. The students should be involved in the learning approach; they must be engaged in the experience.
○ Learning happens everywhere and at any time: open your fantasy and evaluate what may be the best way to reach learning goals.

○    Learning takes care of materials and methods: choose the appropriate combination in order to make learning achievements more effective.

The primary responsibility of the teacher is to create a conducive learning environment by linking knowledge to practical and experiential domains, encouraging active student engagement, and fostering autonomy. Several influential figures have contributed significantly to this pedagogical perspective. Notably, Dewey [12] emphasized the vital role of students' active participation in authentic experiences, while Jerome Bruner [13] introduced the theory of "Learning through discovery". We summarized our goals as follows: to validate the opportunities provided by mandatory remote teaching, activate appropriate formative assessment strategies, and foster the creative potential of students. In the experimental activity, we encouraged students to gain knowledge by engaging in science [14]: observing, questioning, exploring, hypothesizing, testing hypotheses, comparing predictions, and evaluating data. Since education is of vital importance and should connect with lessons experienced during students' daily lives, it is important to encourage education by prompting and encouraging them to work in cooperative groups. In this pedagogical framework, we designed the experimental part inspired by an inquiry-based learning framework [15]. Inquiry-based learning begins with the formulation of a question that is similar to a research question but is presented in a simplified manner [16]. The teacher challenges students to seek the best answers by engaging in individual and group work, thereby fostering a social learning experience influenced by Vygotsky's perspective [17]. To support this process, the teacher can provide practical and theoretical materials, documents, links to relevant resources, and methodological guidance. However, the levels of student involvement and autonomy are crucial. After presenting a phenomenon, the teacher may encourage students to develop their own questions, identify significant variables, and determine a method to explore the relationships among those variables. We can assume that the inquiry-based learning experience is carried out as follows [18]:

○    Questioning: Students generate questions based on a phenomenon, event, or situation presented by the teacher. The teacher can facilitate a brief discussion to collaboratively frame a shared question or initiate small group activities where each group formulates its own question. In certain cases, the teacher may pose a question as a challenge for the entire class.

○    Information research: Students begin exploring the main subject by researching texts or digital resources to locate relevant information. They may also conduct individual or small-group laboratory experiments or observations.

○    Answer preparation: Students interpret the gathered information and work in small groups to synthesize an answer.

○    Discussion and synthesis: Multiple groups that have worked on the same question come together to compare their answers, engage in discussions about differing viewpoints, and strive to reach a summarized shared answer if possible.

○    Reflection and assessment: The researched answers are shared, discussed, and evaluated by the teacher in collaboration with the class. Based on the given answers, the teacher can introduce new content and propose additional questions for further exploration.

Depending on the specific context, the research activity can involve practical experiences or the exploration of organized content to varying degrees. In the experimental part of the study, students were asked to experiment with the subject of acoustic interferometry in the absence of detailed indications and restrictions and with complete autonomy, providing them the freedom to adopt an open approach [19]. High school Physics labs typically follow a closed-ended approach, where the expected outcomes are predetermined, and students simply replicate procedures that are suggested by their teacher. However, as mentioned, we recognized the immense value of granting students the freedom to explore and investigate Physics labs. We came to understand that by allowing students to design and conduct their own investigations, a laboratory setting can offer them diverse and abundant opportunities for making discoveries [20]. At the end of the experiment, each group

had to write a report and prepare a multimedia presentation about their work. We further suggested that all students of each group contribute to the final presentation. Student presentations are a common part of many courses at high school as they are a method that can improve the learning of course material. Increased class interaction and participation are some of the potential benefits of student presentations. Students can also improve their communication and presentation skills by observing their peers' presentations. But, as with any presentation, one of the challenges is to engage the non-presenting students of the remaining groups in the learning experience [21]. Additionally, we have to consider engagement difficulties due to the fact that the experimental works were presented online on a digital platform: Google Meet, in our case. For non-presenting students, we proposed that they utilize the peer evaluation [22] strategy by peer-assessing the presentations of each group during their own session [23]. In the assessment process, students are not merely passive participants who provide an assessment of performance. Instead, they can assume active roles of significant importance in terms of their own learning. Indeed, students can actively participate in the assessment process [24] by incorporating criteria and indicators into their own assignments (self-assessment) and evaluating the work of their peers (peer assessment). As early as 1986, Weaver and Cotrell [25] emphasized the following: "Peer assessment stresses competencies, encourages involvement, promotes attention to learning, establishes a clearer framework of reference, promotes excellence, increases feedback, and enhances participation as well as the teacher's accountability". Finally, the teachers evaluated the presentations by utilizing the same rubric that students used in the peer assessment.

## 2. Materials and Methods

The project was performed in Liceo Copernico in Brescia and Liceo Vittorio Veneto in Milano during the 2020–2021 school year and in Liceo Vittorio Veneto during the 2021–2022 school year; these are named Event 1, Event 2, and Event 3, respectively. The activity was supervised by a local school coordinator in addition to 3 faculty members and 2 tutors that were appointed by our university. Next, we named the group composed of the 2 tutors and 1 faculty professor "teachers". The project was devised according to the following structure, as shown in Figure 1, which we briefly describe.

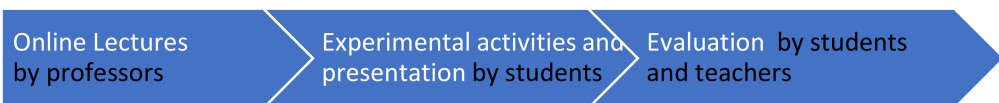

**Figure 1.** Main features of the project (PCTO).

- Online lectures and webinars by the faculty professors: In the first part of the project, the students attended four webinars presented by university professors about different topics within the field of wave phenomena. The first two lectures were focused on introducing the basic aspects of wave interaction ("Wave Physics in everyday life"), including topics that were already included in school programs (acoustic and electromagnetic waves, interference, and diffraction) and some hints for more advanced concepts (e.g., wave/particle duality). The third and fourth webinars were dedicated to the concept of modeling a physical phenomenon—including comparisons among the approaches of different global cultures—and provided a practical description of scientific experiments involving antimatter ("The foundational structures of Western science" and "Antimatter interferometry").
- Experimental activities developed and performed by the student groups: The students formed groups with 3, 4, or 5 people each, based on their own criteria. They had the task of planning/carrying out an experiment on the subject of acoustic interferometry in the absence of detailed indications and restrictions, with complete autonomy and the freedom to adopt a creative approach. The groups were given a total period of 4 months for this phase of the activity, during which they could rely on the support

of university tutors and school supervisors. At the end of the agreed time period, each group had to write a report and prepare a multimedia presentation about their experiment. The COVID restrictions at that time only applied to public places such as schools, universities, and large gathering venues, so students could handle the experiments in groups in their homes without any restrictions. In each event, five groups were formed, for a total of 15 groups.

- Presentation and reports of each group activity: Each group presented the experimental work and evaluated the group work presentation of other classmates with respect to each session and event. Teachers also evaluated the reports and presentations of each group during the 3 different sessions (Event 1, Event 2, and Event 3). The rubric consisted of 8 questions with a 4-point Likert scale ("not at all" (1), "slightly" (2), "enough" (3), and "very much so" (4)), investigating different features of the presentation: the layout, the introduction, the description of the experiment, the data, data analysis, the conclusion, the exposition of the presenters, and an overall rating. This was a type of formative assessment based on two important features: the rapidity and informality of the assessment. After the completion of all the students' presentations, the reference teacher proceeded to inform each group about their score evaluations. Finally, we proposed a survey concerning the entire project, from lectures and experimental activities to peer evaluations. The questionnaire was co-designed through progressive interactions with the teachers, with the contribution of expert consultations. This survey comprised 17 questions: 6 were related to the webinars hosted by faculty professors (part A), 4 were related to experimental activities (part B), 4 were related to the evaluation process (part C), and 3 were related to the overall project PCTO. Question 13, in particular, regarded the entire perception of the project. The questions were grouped in different sections, as in Figure 1, for clearance purposes, and we could not identify any latent variables in each part (Supplementary Materials). The scale used was the same as the scale that was used in the evaluation rubric, which is a 4-point Likert scale. All students participated in the presentation evaluation session, while not all students filled in the feedback questionnaire. This paper focuses on discussing the designed strategies and the possibilities of exploiting one of these engaging strategies, which is peer evaluation. In this context, we developed the following research questions:

  ○ Is this student-centered approach appreciated by high school students?
  ○ In a formative context, is peer evaluation comparable with the evaluation provided by teachers?

First, we collected information about the topics that the learners dealt with in the experiments and how the students developed the activities; then, we analyzed the data concerning the evaluation and the survey. We initially performed a descriptive analysis of the feedback questionnaire and evaluation data. Afterward, being aware of the different sample sizes of the evaluators (47 students vs. 4 teachers), we investigated the difference in teacher–student evaluations of the presentations by using inferential statistics.

## 3. Results

Most groups (11 out of 15) chose to investigate the phenomenon of interference between soundwaves, taking cues from the description of Young's experiment, which was presented during one of the introductory webinars. The remaining groups chose to study the acoustic resonance of beats and the Doppler effect. The complexity of such experiments, enhanced by the need to conduct them at home due to pandemic-induced restrictions, made it particularly difficult to obtain experimental results that are in agreement with expectations. As an example, accounting for reverberation in a closed environment as well as challenges in obtaining wavelength monochromaticity and coherence values, were critical issues that forced the students to wonder about the difference between analyzing actual experimental results and studying the predictions of simplified theoretical models. Such questions were widely presented and discussed in the experimental reports as well

as the presentations they gave. Another observation that can be remarked upon is the decision of every group to employ dedicated software and apps to generate, record, and measure the physical properties of soundwaves; this is a confirmation of the ever-growing integration of multimedia technology in experimental teaching. Specifically, the following apps were used: Physics Toolbox Suite, SpectrumView, Spectroid, Soundcorset, Sonic Visualizer, Online Tone Generator, and Audacity. At the end of this phase, the teachers rated the experimental reports of each group based on a common evaluation grid, in which they assessed different aspects of the report with a score from 1 to 4; these data were not analyzed in this paper. Simultaneously, the presentations were proposed on the Google Meet platform and were evaluated by the teachers and students. As described in the Introduction, we collected data from three different samples: two classes in 2021, named Event 1 and Event 2, and one class in 2022, named Event 3. We analyzed the Likert scale data, assigning scores from 1 to 4 to the answers, as described in Materials and Methods. The results are reported in two subsections, and they are related to the feedback provided for the entire project, which is the survey and the peer evaluations of the presentations.

### 3.1. The Survey

In this study, 30 students out of 45 filled out the survey concerning PCTO 2021, while 12 of 16 students filled out the survey related to PCTO 2022; that is, 69% of the students responded to the survey on a voluntary basis. In 2021, we analyzed the data from PCTO 2021 and decided to propose the same project for the 2021–2022 school year, named Event 3, as the feedback provided by students was that the project was clearly helpful [26]. Furthermore, in 2021–2022, even when COVID-19 restrictions were lifted, the scholar institute still did not allow students to attend the project in person. In Figure 2, the percentage frequency distribution that was collected in the questionnaire is shown, emphasizing the results for each part of the questionnaire and question 13. From frequency analyses, we deduced that more than 85% of students scored "3" or "4" on a scale from 1 to 4 for part B concerning the experimental and group activity. Furthermore, we noticed that 97% of students appreciated the evaluation strategies, assigning "3" or "4" in part C, as shown in the graphs. Finally, question 13 obtained a score of "3" or "4", as scored by more than 90% of students. Still using a descriptive approach, Figure 3 indicates the mean values, and we can observe that the students appreciated the entire project, as the value is clearly greater than "3".

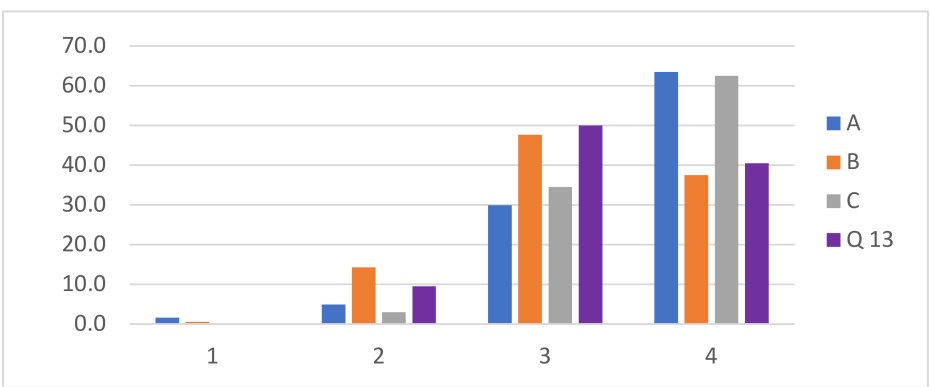

**Figure 2.** Frequency percentage distribution in the questionnaire: parts A, B, and C and Question 13.

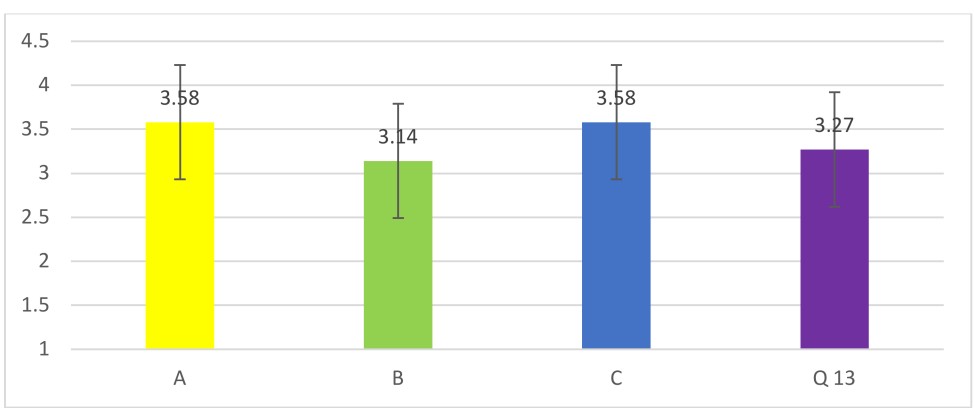

**Figure 3.** Mean value in the questionnaire: parts A, B, and C and question 13.

*3.2. Peer Evaluation*

We collected 360 evaluations presented by the three teachers and 1834 evaluations provided by the total number of students involved in the project. We performed three different types of descriptive analysis, from the entire sample to the single group, with the first one involving the entire sample regardless of the event and the group. The second one took into account the event, while in the third one, we compared the score obtained in each group. By using Excel and SPSS statistical software, we first computed the coefficient of skewness and kurtosis to check the asymmetry and peakedness of these two distributions. In any case, due to a large amount of data and referring to the central limit theorem, we can assume a normal distribution for the two different samples involved in this first analysis: the evaluations of the presentations carried out by the teachers and the evaluations provided by the students regardless of the groups and schools. First, we obtained the normalized frequency distribution percentage and the descriptive statistics, as shown in Table 1 and Figure 4. The bar graph in Figure 4 indicates the percentage of the frequency distribution with respect to students and teachers, referring to the total amount of data.

**Table 1.** Descriptive statistics scores for student and teacher evaluations about presentations carried out by groups.

| Students | | Teachers | |
|---|---|---|---|
| Mean value | 3.2 | Mean value | 2.6 |
| Median | 3.0 | Median | 3.0 |
| Mode | 3.0 | Mode | 3.0 |
| Standard Deviation | 0.6 | Standard Deviation | 0.8 |
| Variance | 0.4 | Variance | 0.6 |

We can observe in Figure 4 that a score of "2" and "4" showed opposite distributions; 33% of evaluations provided by students scored "4", but "4" only comprised 10.6% of the teachers' evaluations. Furthermore, score "3" exhibited a difference of ten percentage points. Finally, in order to check the reliability of this "shift", we used a Student's *t*-test comparison for samples exhibiting different variances [27]. The conditions for using Student's *t*-test were guaranteed in this case by the central limit theorem. However, as the samples had unequal variances, we performed a Welch's *t*-test. We verified the consistency of the shift observed in the frequency distribution by analyzing the difference in the mean value. The *t*-test shows that the difference was significant from a statistical point of view, and the probability that this difference had not been due to a coincidence was clearly greater than 99.9%, as reported in Table 2, where the results are summarized.

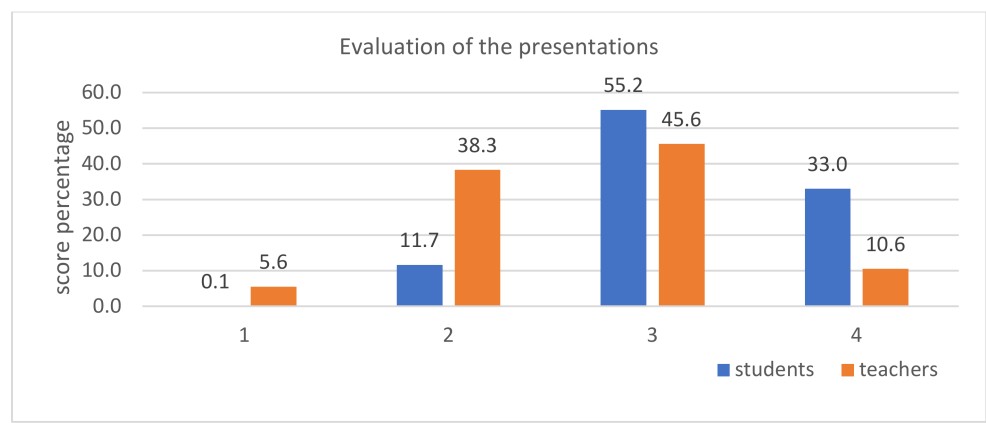

**Figure 4.** Frequency percentage distribution in presentation evaluations, divided by students and teachers.

**Table 2.** Results of the *t*-test comparing student and teacher evaluations of the presentations.

|  | N° Answers | Mean Value | St. Dev. | *t*-Test *p*-Value | *t*-Test Cohen D |
|---|---|---|---|---|---|
| Students | 1834 | 3.2 | 0.6 | *p* << 0.001 | 0.66 |
| Teachers | 360 | 2.6 | 0.7 | | |

In order to strengthen this result, we checked if this variation was sufficiently large from a statistical point of view by computing the effect size coefficient, Cohen's D, which is equal to 0.66 [27]. In the literature, Cohen [28] suggests that d = 0.50 indicates a medium effect size and d = 0.80 is a large effect size in the Student's *t*-test.

In order to expand the analysis, we divided the larger sample into three different events, and we investigated the outcomes of each. After computing the mean value, we checked the consistency of the differences with respect to the mean value by conducting a *t*-test. The results are summarized in Table 3.

**Table 3.** Results of the *t*-test comparing student and teacher evaluations of the presentations, referring to each event.

|  |  | N of Answers | Mean Value | St. Dev. | *t*-Test *p*-Value | *t*-Test Cohen D |
|---|---|---|---|---|---|---|
| Event1 | Students | 712 | 3.2 | 0.6 | *p* << 0.001 | 0.66 |
|  | Teachers | 120 | 2.9 | 0.7 | *p* << 0.001 | |
| Event2 | Students | 612 | 3.2 | 0.6 | *p* << 0.001 | 0.63 |
|  | Teachers | 120 | 2.6 | 0.7 | *p* << 0.001 | |
| Event3 | Students | 510 | 3.2 | 0.7 | *p* << 0.001 | 0.67 |
|  | Teachers | 120 | 2.6 | 0.7 | *p* << 0.001 | |

Table 3 shows that the difference in the mean value for each event was not a result of coincidence, and the difference exhibited a probability greater than 99%. In order to strengthen this result, we checked whether this variation was large enough from a statistical point of view by computing the effect size coefficient, Cohen's D, as shown in the table. Finally, we performed descriptive statistics by reducing the large sample into the 15 groups in which the students worked together. For each group, the frequency distribution and the mean value comparing teachers and students are reported in Table 4.

**Table 4.** Percentage frequency distribution for each group concerning teacher and student evaluations of the presentations.

| | Group | Score | Student's Score Percentage | Teachers' Score Percentage | Mean Value Students | Mean Value Teachers |
|---|---|---|---|---|---|---|
| *Event 1* | 1 | 1 | 0 | 0 | 3.5 | 3.4 |
| | | 2 | 5.1 | 4.2 | | |
| | | 3 | 43.4 | 50 | | |
| | | 4 | 51.5 | 45.8 | | |
| | 2 | 1 | 0 | 0 | 3.0 | 2.6 |
| | | 2 | 5.1 | 4.2 | | |
| | | 3 | 43.4 | 50 | | |
| | | 4 | 51.5 | 45.8 | | |
| | 3 | 1 | 0 | 8.3 | 3.2 | 2.6 |
| | | 2 | 11.8 | 29.2 | | |
| | | 3 | 56.9 | 58.3 | | |
| | | 4 | 31.3 | 4.2 | | |
| | 4 | 1 | 0 | 4.2 | 3.3 | 2.4 |
| | | 2 | 8.3 | 58.3 | | |
| | | 3 | 54.9 | 29.2 | | |
| | | 4 | 36.8 | 8.3 | | |
| | 5 | 1 | 0 | 12.5 | 3.0 | 2.5 |
| | | 2 | 17.4 | 29.2 | | |
| | | 3 | 66 | 58.3 | | |
| | | 4 | 16.7 | 0 | | |
| *Event 2* | 6 | 1 | 0 | 8.3 | 3.4 | 2.5 |
| | | 2 | 7.5 | 41.7 | | |
| | | 3 | 47.5 | 37.5 | | |
| | | 4 | 45 | 12.5 | | |
| | 7 | 1 | 0 | 0 | 3.2 | 2.7 |
| | | 2 | 10.9 | 41.7 | | |
| | | 3 | 54.7 | 50 | | |
| | | 4 | 34.4 | 8.3 | | |
| | 8 | 1 | 0 | 12.5 | 3.1 | 2.2 |
| | | 2 | 10.1 | 54.2 | | |
| | | 3 | 70.6 | 33.3 | | |
| | | 4 | 19.3 | 0 | | |
| | 9 | 1 | 0 | 4.2 | 3.2 | 2.9 |
| | | 2 | 10.9 | 16.7 | | |
| | | 3 | 54.6 | 62.5 | | |
| | | 4 | 34.5 | 16.7 | | |
| | 10 | 1 | 0 | 4.2 | 3.3 | 2.5 |
| | | 2 | 8.7 | 45.8 | | |
| | | 3 | 55.6 | 45.8 | | |
| | | 4 | 35.7 | 4.2 | | |

**Table 4.** *Cont.*

| | Group | Score | Student's Score Percentage | Teachers' Score Percentage | Mean Value Students | Mean Value Teachers |
|---|---|---|---|---|---|---|
| *E v e n t 3* | 11 | 1 | 0 | 8.3 | 3.2 | 2.5 |
| | | 2 | 10.6 | 41.7 | | |
| | | 3 | 56.7 | 37.5 | | |
| | | 4 | 32.7 | 12.5 | | |
| | 12 | 1 | 0 | 0 | 3.1 | 2.7 |
| | | 2 | 17.1 | 41.7 | | |
| | | 3 | 58.6 | 50 | | |
| | | 4 | 24.3 | 8.3 | | |
| | 13 | 1 | 0 | 12.5 | 3.4 | 2.2 |
| | | 2 | 10.6 | 54.2 | | |
| | | 3 | 41.3 | 33.3 | | |
| | | 4 | 48.1 | 0 | | |
| | 14 | 1 | 0 | 4.2 | 3.4 | 3.0 |
| | | 2 | 8.3 | 16.7 | | |
| | | 3 | 41.7 | 62.5 | | |
| | | 4 | 50 | 16.7 | | |
| | 15 | 1 | 0 | 4.2 | 3.1 | 2.5 |
| | | 2 | 16.8 | 45.8 | | |
| | | 3 | 60 | 45.8 | | |
| | | 4 | 23.2 | 4.2 | | |

First, we highlighted that the mean value of the score given by students is higher than the mean value of the teachers' evaluations. Furthermore, if we consider the frequency distribution, we can observe the exchange in the score's distribution. The percentage of scores "1 and 2" is greater in the teachers' column, with the exception of groups 1 and 2, while the percentage of score "4" is greater in the students' column for each group.

## 4. Discussion and Conclusions

In 2020–2021, restrictions arising from the COVID-19 pandemic introduced a complete renewal of Physics education projects dedicated to students at the end of high school. The activity was a PCTO (vocational training during high school) project, which challenged students to develop and perform an experiment related to acoustic interferometry after attending some introductory webinars, to write an experimental report and prepare a presentation, and to participate in the peer-to-peer evaluation. As the project proposed during the 2020–2021 school year was clearly appreciated by students, the same project was repeated during the 2021–2022 school year. The project proposed a student-centered approach, involving students in setting up and performing an experimental activity and peer evaluation. The methods and strategies used to design the entire project refer to a statement that we strongly support, which is "learning is an experience", referring to the pedagogical constructivist approach [29]. The results in Reference [30] corroborate the assertion that incorporating open-ended activities can potentially yield favorable outcomes with respect to the student's understanding of the nature of experimental Physics, as well as their emotional state and self-assurance during Physics experiments.

Students appreciated the entire project: the webinars, experimental activities, and peer evaluation. Even if the experimental work was more demanding than the webinars, it

was well-valued. Furthermore, students highly regarded the fact that they could evaluate their classmates, which comprised the peer evaluation process. We can surely discuss the link between students' high opinions on a teaching strategy and effective learning [31]. Unfortunately, we could not collect any other information about the future curricula and the future performance of any students.

The data analysis showed us that the presentations were evaluated by students and teachers using different scores, and this difference is significant from a statistical point of view; students rewarded scores more than teachers. In the literature, the discussion about the use and reliability of peer evaluation as a formative assessment [32] can be considered relevant from the point of view of the student's learning experience, as observed in their evaluation of the activity. The accuracy of peer evaluation as an instrument for summative assessments, potentially considered for final grading, still remains a big question that needs to be further examined [33]. Different strategies for improving these methods have been considered in the literature [22]. In the context of this research study, we might consider a combination of peer evaluation and self-evaluation as a more reliable approach [34]. Additionally, exposure to peer learning and peer evaluation instruments and procedures needs to be considered as part of the pedagogical design [35]. Our results may not allow us to infer strong opinions because of two limitations; the sample, at first, was non-probabilistic. The second limitation is the choice not to deepen the analysis of the peer evaluation data with stronger statistics tests (i.e., Kendall's tau test). Moreover, future research could deepen these results [36] by checking the type of internal consistency in each group of evaluators.

This research study could also be expanded by collecting more data, which means proposing the same project to a larger population that involves more students and teachers, although there is an intrinsic limitation with respect to deciding on which components to use in comparisons. Is it better to compare a large amount of data, as we have carried out in this study, regardless of the group, or is it more reliable to carry out a comparison between the rates indicated by the same group? In some sense, we are asking whether the assessment methods are reliable regardless of the components that are evaluated.

**Supplementary Materials:** The following supporting information can be downloaded at: https://www.mdpi.com/article/10.3390/educsci13080799/s1.

**Author Contributions:** Conceptualization, M.Z.; formal analysis, R.L.M.; investigation, R.L.M. and M.B.; writing—original draft preparation, R.L.M.; writing—review and editing, P.G. and M.B.; supervision, M.Z. and J.E.R.; project administration, M.Z.; All authors have read and agreed to the published version of the manuscript.

**Funding:** This research received no external funding.

**Institutional Review Board Statement:** The study was conducted in accordance with the Declaration of Helsinki and approved by the Institutional Review Board of Politecnico di Milano.

**Informed Consent Statement:** Informed consent was obtained from all subjects involved in the study.

**Data Availability Statement:** Datasets are available on request from the authors.

**Conflicts of Interest:** The authors declare no conflict of interest.

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
