# Peer review of "Exploring Effective Physics Teaching Strategies in High Schools during the COVID-19 Pandemic"

_education, doi:10.3390/educsci13080799_

Round 1

Reviewer 1 Report

The manuscript deals with an interesting issue. Students’ opinions about proposed vocational training during high school and peer evaluation within the project are discussed. However, the authors should make an effort to improve the quality of their manuscript. The article is not clear, cohesive and logical. The importance of this research in the international context should be emphasized.

Based on the title (“Novel physics teaching strategies in high schools and universities during the COVID-19 pandemic”) readers can expect more than the paper offers. It would be better to write a more accurate and concrete title – it should predict the content of the paper. Besides, strategies for teaching university students were not discussed.

In accordance with the Instructions for Authors, the authors should add in abstract: “3) Results: Summarize the article's main findings; and 4) Conclusion: Indicate the main conclusions or interpretations.”

lines 5-7, lines 25-30...

“The delicate transition between high school and university has become a key point to focus on, leading many institutions to replan projects dedicated to students involved in this transition.” – authors didn’t discuss this issue in regard to their research.  

lines 13-14

“The second peculiarity is creativity with respect to how students handle experimental activities.”

“Creativity” is mentioned only in the abstract. This concept is not introduced and there are no data about students’ creativity within this research.

lines 87-105

There are different types of inquiry-based learning...

lines 38-45

During their last three years of high school, students are employed in a vocational training program at different public or private institutions, factories, research centers, firms, universities, etc.( – this is regarding PCTO?) This activity, starting from a series of webinars about optics and about different perspectives on the experimental method, ultimately led different groups of students(three or four students for each group) in devising an acoustic interferometry experiment from scratch; finally, they presented their work and evaluated the presentations of their classmates. (- what about this? Is this about this research? How is this correlated with PCTO? Should be explained...)

line 167

How were the reports and presentations evaluated? Was that anonymized or the students knew about the outcome of their peer evaluation?

lines 212-213

Simultaneously, the presentations were proposed on the Google Meet platform and were evaluated by the teachers and students.

but lines 125-126:

Additionally, we have to consider engagement difficulties due to the fact that the experimental works were presented online on a digital platform: Zoom platform in our case.

What was used? Zoom or Google Meet?

line 237 - 3.2. Peer evaluation

The analysis was carried out about “the evaluations of the presentations carried out by the teachers and the evaluations provided by the students regardless of the groups and schools” - This needs a very good explanation. Without good justification the current analysis seems incorrect. In the current analysis students could evaluate one group with 4 and teachers could evaluate the same group with one, for another group it could be: students’ evaluation 1, teachers’ evaluation 4... so it can be concluded that students’ evaluation is in accordance with teachers’ evaluation (this way we cannot discuss similarities between students’ evaluation and teachers’ evaluation)

* Further text makes this clear since different data are presented but I still suggest that authors explain this at the beginning of the section Peer evaluation

The answers on research questions are not clearly stated nor discussed enough.

1. “Is this student-centered approach suitable for high school students?” – Regarding this research results how is this answered? ... “Students appreciated the entire project: the webinars, experimental activities. and peer evaluation. Even if the experimental work was more demanding than the webinars, it was well valued. Furthermore, students highly regarded the fact that they could evaluate their classmates, which comprised the peer evaluation process.” but was it useful, did students learn more in regard to traditional learning, did they learn easier, faster... Students appreciating it is not enough to conclude about its suitability...

2. “In a formative context, is peer evaluation comparable with the evaluation provided by teachers?” – The authors should additionally discuss this question based on their own results. What is authors’ opinion about the answer to this question? The answer on this question in the Discussion section of this version of the article is mainly based on cited literature instead of this research results. (The use of literature is OK but part about this research should be discussed further.)

How was a PCTO (vocational training during high school) project related to students’ school activities? Was the outcome of evaluation in PCTO influencing evaluation of students’ physics performance in school? How were students motivated to participate in this research?

How was the group work carried out? It seems unlikely that students within groups met in person during COVID-19 restrictions but how it was group work if each student was doing experimental work at his/her own home?

Research sample should be described in more detail. Which method of sampling was used? Were both genders equally represented? ...

The research instrument should be given in the appendix since it is not described well enough.

There is no information about the reliability and validity of the questionnaire – should be added.

References in the Reference list are not described in accordance with the Instructions for Authors.

Reviewer 2 Report

Overall, I find the article interesting and an original contribution to research and the expansion of teaching methodologies.

Below are some notes/thoughts that the authors might consider:

In general, the Introduction should be reorganised, in my opinion, into chapters in which one can distinguish 1) the research already carried out in this field 2) some general aspects of the research carried out and 3) the physics involved (e.g. a brief description of acoustic interferometry and a reference on what the 'Young experiment' is - because in several school books the latter is described referring to water waves, and not acoustic waves), and considerations as to why precisely these aspects were deemed suitable for distance learning 4) the objectives of the research and the organisation of the article.

Lines 26-28: high school is presented as a transitional path. True, but I would also add a value in itself of cultural formation and, through this, of the person.

Lines 33-40: the PCTO should be presented a little more precisely, e.g. by mentioning the law establishing it and some further basic features, e.g. compulsoriness and number of hours required.

Lines 40-41: the authors move from the general discourse to the particular topic dealt with in the article, but does so without signalling it enough ("This activity" seems to refer to the PCTO in general, and not to the one proposed in the following).

Lines 50-53: I would add that learning also takes place through a maturation and refinement of one's conceptual system and the language that conveys it (e.g. use of metaphors, imagery...) (perhaps some more references can be made to "prominent figures").

line 59. Just a consideration of mine, which the authors do not need to take into account, but is only to stimulate a possible debate in the scientific and pedagogical community. Are we sure that the classic 'frontal lecture' is passive? If it is, is theatre, music, cinema also passive? Or are they activities, on the other hand, which, precisely when done well, are able to stimulate active participation, i.e. move the learner (or spectator) to reflect? In my opinion, we are forgetting that a good teacher knows how to lead to learning even when giving a frontal lecture, if he or she knows how to do it properly, that is, using mental images and language that foster the development of ideas in learners.

Line 137: it is not clear to me, who the "collaborators" are.

Line 148: what was the need to involve university teachers? weren't school teachers enough? Maybe this has something to do with the fact that the PCTO route has to be in collaboration with organisations and companies; if so, I think this needs to be said here.

Line 154: I do not understand what is meant by 'approaches of different global cultures' (e.g. science vs. philosophy? science vs. religion? science vs. superstition?).

About the section "Materials and methods": why not include the materials used by the students? Did they use any tools, measurement instrument etc.?

Line 221: is there any particular reason why as many as 1/3 of the students did not complete the questionnaire? seems like a high number.

Tables 1, 2 and 3: if the standard deviation already concerns the first decimal place, does it make sense to specify the subsequent decimal places in the average? Usually a quantity such as: 3.21 +/- 0.64 is rendered as: 3.2 +/- 0.6. The further decimal digits are not meaningful, i.e. they do not add anything, but rather make the message unnecessarily complicated. I suggest modifying the numbers in the table in light of this.

Not bad.

Round 2

Reviewer 1 Report

I believe the manuscript has been sufficiently improved and can be accepted for publication in present form.

Author Response

As suggested by the respected Reviewer, we thank for being accepted.

Reviewer 2 Report

I keep thinking that the Introduction could be written better, perhaps also shortened and made more efficient.

Should be improved. Punctuation needs to be revised. 

Author Response

As suggested by the respected Reviewer, we modified the paper modifying the introduction and improving the English Language